# Genome-Wide Association Study of Body Conformation Traits in Tashi Goats (*Capra hircus*)

**DOI:** 10.3390/ani14081145

**Published:** 2024-04-09

**Authors:** Rong Yang, Di Zhou, Xiaoshan Tan, Zhonghai Zhao, Yanli Lv, Xingzhou Tian, Liqun Ren, Yan Wang, Jun Li, Yongju Zhao, Jipan Zhang

**Affiliations:** 1Guizhou Provincial Breeding Livestock and Poultry Germplasm Determination Center, Guiyang 550018, China; 15761629346@163.com (R.Y.); l18385176795@163.com (Y.L.);; 2Zunyi Animal Husbandry and Fishery Station, Zunyi 563000, China; 3College of Animal Science, Guizhou University, Guiyang 550025, China; 4College of Animal Science and Technology, Southwest University, Chongqing 400715, China; zyongju@163.com

**Keywords:** goats, GWAS, SNP, body conformation

## Abstract

**Simple Summary:**

To explore the genomic variations associated with goats’ body conformation traits, we performed a genome-wide association study (GWAS) on Tashi goats. We assessed eight body conformation traits in 155 Tashi goats and performed whole-genome sequencing on 100 of them. We obtained 1676.4 Gb of raw data and identified 11,257,923 qualified single nucleotide polymorphisms (SNPs). GWAS revealed 109, 20, 52, 14, 62, 51, 70, and 7 SNPs significantly associated with body height, body length, chest depth, chest width, chest girth, rump width, rump height, and cannon bone circumference, respectively. We annotated 183 genes based on the significant SNPs’ physical locations. Notably, several SNPs have been identified multiple times in two regions, chr.10:25988403-26102739 and chr.11:88216493-89250659, where candidate genes such as *FNTB*, *CHURC1*, and *RNF144* could be crucial for goat body conformation traits. Our findings offer significant insights into body conformation and support the use of molecular breeding in meat goats.

**Abstract:**

Identifying genetic markers of economically valuable traits has practical benefits for the meat goat industry. To better understand the genomic variations influencing body conformation traits, a genome-wide association study was performed on Tashi goats, an indigenous Chinese goat breed. A total of 155 Tashi goats were phenotyped for eight body conformation traits: body height, body length, chest depth, chest width, chest girth, rump width, rump height, and cannon bone circumference. Then, 100 Tashi goats were randomly selected for whole-genome sequencing and genotyped. We obtained 1676.4 Gb of raw data with an average sequencing depth of 6.2X. Clean reads were aligned to the ARS1.2 reference genome, and 11,257,923 single nucleotide polymorphisms (SNPs) were identified. The structure analysis showed that these Tashi goats were almost not genetically related. The 109, 20, 52, 14, 62, 51, 70, and 7 SNPs were significantly associated with body height, body length, chest depth, chest width, chest girth, rump width, rump height, and cannon bone circumference. Within the ±500 kb region of significant SNPs, 183 genes were annotated. The most significantly enriched KEGG pathway was “olfactory transduction”, and the most significantly enriched gene ontology (GO) terms were “cellular process”, “cellular anatomical entity”, and “molecular transducer activity”. Interestingly, we found several SNPs on chromosomes 10 and 11 that have been identified multiple times for all eight body conformation traits located in two fragments (114 kb and 1.03 Mb). In chr.10:25988403-26102739, the six SNPs were tightly linked, the TACTAG genotype was the highest at 91.8%, and the *FNTB* (Farnesyltransferase, CAAX Box Beta) and *CHURC1* (Churchill Domain Containing 1) genes were located. In chr.11:88216493-89250659, ten SNPs were identified with several dependent linkage disequilibrium (LD) blocks, and seven related genes were annotated, but no significant SNP was located in them. Our results provide valuable biological information for improving growth performance with practical applications for genomic selection in goats.

## 1. Introduction

As ruminants, goats play essential roles by producing milk, meat, mohair, cashmere, and furs for humans [1]. Goats are well adapted to harsh environments, such as deserts, mountains, and cold and tropical regions, so they are widely distributed and contribute to the economic resources of developing countries, especially in Southeast Asia and Africa [2]. Despite the escalating global demand for other goat-derived products, such as cashmere and milk, meat is still the main product of goat farming. The pursuit of increased productivity drives the critical need to unravel the genetic determinants governing economically important traits in goats, prompting the adoption of innovative methodologies [3,4].Breeding goats with excellent growth traits using molecular breeding technologies (such as genomic selection) has become essential.

The body shape traits of a goat are the most direct production indexes, which directly affect prolificacy traits [5] or slaughter traits, such as carcass mass, net meat mass, and slaughter rate [6]. In scientific research and production activity, the body conformation traits that are most frequently used include body height, body length, chest depth, chest width, chest girth, rump width, rump height, and cannon bone circumference. These traits are polygenic and highly influenced by environmental factors, so it is difficult to discover associated genes [7]. Recently, methods based on bioinformatics (genome and transcriptome analyses) have been applied to study the genetic basis of body conformation traits. Luigi-Serra et al. [8] genotyped Murciano-Granadina goats using the GoatSNP50 BeadChip and identified some variations associated with body, udder, and leg conformation traits. The deletion variants within the PRDM6 goat gene affected growth traits such as cannon bone circumference, chest depth, and chest width in the early growth stage [9]. In Hainan black goats, the paired-such as homeodomain 2 (*PITX2*) was significantly associated with body height and body length [10]. In Shanbei white cashmere goats, two indels in the prolactin receptor (*PRLR*) gene were significantly correlated with body length, body height, chest depth, heart girth, and cannon bone circumference [11].

The genome-wide association study (GWAS) technique is the most popular method for analyzing associations between genetic variations and complex traits [12,13]. The effect size of genetic variations, such as single nucleotide polymorphisms (SNPs), was calculated based on linkage disequilibrium (LD). Genotyping methodologies include microarray, whole genome sequencing, and simplified genome sequencing. Although genome resequencing is more expensive than other methodologies, it obtains more numerous and more effective SNPs. The GWAS technique has enabled the identification of genes linked to various traits in goats, such as milk quality [14,15], meat quality (fat and protein content) [16], growth traits (weaning mass) [17,18], and body conformation traits (body height and length) [19,20]. The genetic insights gleaned from these studies not only present promising avenues for selective breeding but also pave the way for sustainable enhancements in productive performance.

The Tashi goat is a local goat breed, mainly distributed in the Rongjiang and Leishan counties, Guizhou Province, China. However, few relevant studies on this breed have been published. In this study, we investigated 155 Tashi goats and performed whole-genome sequencing on 100 of them. Then, we performed a population structure analysis and GWAS to identify genes associated with body composition traits.

## 2. Materials and Methods

### 2.1. Animals and Phenotyping

This study was approved by the Guizhou University Experimental Animal Ethics Committee (No. EAE-GZU-2023-E047). The Tashi goats (Figure 1A) used in this study were obtained from different villages in Rongjiang County, Guizhou Province, China. On dozens of farms, adult, healthy animals were selected, and no direct kinship was found among them. The investigative work was performed by a survey team comprising four experimenters: one person kept the goat calm, one measured the body conformation traits, one collected 3–5 mL of blood from the goat’s jugular vein, and another recorded the data. The blood samples were kept in an ice box, transferred to the laboratory, and stored at −80 °C until subsequent use.

Following previous studies [8,21], eight body conformation traits were measured in 155 Tashi goats.

(1)Body height (BH) was measured from the ground to the top of the withers.(2)Body length (BL) was measured from the point of the shoulder to the pin bone.(3)Chest depth (CD) was measured as the distance between the top of the spine and the bottom of the body at the beginning of the last rib.(4)Chest width (CW) was measured as the inside surface of the chest between the tops of the front legs.(5)Chest girth (CG) was measured as the chest circumference behind the shoulder blades in a vertical plane.(6)Rump width (RW) was measured as the distance between the most posterior points of the pin bones (ischial tuberosities).(7)Rump height (RH) was measured from the ground to the top of the pin bones.(8)Cannon bone circumference (CC) was measured as the horizontal circumference at the thinnest point of the left forelimb.

### 2.2. Genome Sequencing and Genotyping

Of the above-mentioned 155 samples, 100 blood samples (limited to research costs) were randomly selected for DNA extraction using the standard phenol-chloroform protocol. The concentration, integrity, and purity of the genomic DNA were assessed using agarose gel electrophoresis and a NanoDrop spectrophotometer (Thermo Scientific, Waltham, MA, USA). The qualified DNA samples were randomly broken into fragments of about 150 bp. After construction, the DNA library was sequenced by Biomaker (Beijing, China) using the DNBSEQ-T7 platform.

After obtaining the original off-machine data, we preprocessed the raw reads using the fastp tool (version: 0.23.4, [22]) to obtain clean reads. Whole-genome sequencing data were further processed using Sentieon Genomics software (version: sentieon-genomics-202308, [23]) to implement alignment and variant detection. In brief, the clean reads were aligned to the reference genome (*Capra hircus* ARS1.2, [24]) using bwa software (version: 0.7.17, [25]). The bam files were sorted, duplicates were marked, and an index was created; then the genomic variant call format (gVCF) file was obtained by using the Sentieon haplotyper module. Joint variant calling was performed on all gVCF files using the Sentieon GVCFtyper module. The GATK (version: 4.1.8.1, [26]) SelectVariants module was used to split the SNP variates. The SNPs were pruned if minor allele frequency (MAF) was <0.05, missingness per individual was >40%, or missingness per marker was >10%, while individuals were excluded if their genotyping rate was <90%. The qualified SNPs and individuals were retained for subsequent analysis. A SNP density plot was drawn using the R package CMplot (version: 4.5.0, [27]).

### 2.3. Population Structure Analysis

To understand the basic population structure and individual variation, a principal component analysis (PCA), relatedness analysis, and population genetic structure analysis were conducted. The PCA was performed using Plink (version: 0.76, [28]), and the first three PCs were visualized using the R package ggplot2 (version: 3.4.4). A kinship matrix (G matrix) was calculated using GEMMA software (version: 0.98.5, [29]), and the relatedness among individuals was visualized using the R package heatmap (version: 1.0.12). The hypothetical number of the subpopulation (K) was set to 1–7, the Admixture (version: 1.3.0, [30]) software was run, and a cross-validation error (cv) was calculated. Population structure results (Q files) were visualized using the Python package PONG (version: 1.5, [31]).

### 2.4. Genome-Wide Association Studies

The first five PCs and the kinship matrix were incorporated in the single-locus GWAS using GEMMA [29]. Linear mixed models (LMMs) were conducted as described previously [32], using the following formula:y = Xβ + Zu + e
where y is the morphological trait being analyzed, β is a matrix of covariates (fixed effects, such as sex and age), u is the vector of random effects and follows a normal distribution, and e is the vector of residual effects. To test the null hypothesis, β = 0 for each SNP, and the likelihood ratio test and the Wald test were executed using GEMMA. The GWAS was performed on every trait, and *p* < 1 × 10^−6^ was selected as the significance threshold for SNPs. To visualize the results, Manhattan plots and quantile-quantile plots (QQ plots) were drawn using R packages, including CMplot [27] and qqman [33].

### 2.5. Functional Enrichment Analysis

For SNPs associated with different body conformation traits, we identified SNPs that occurred at high frequencies. LD between SNP markers was calculated using the jar package Haploview (version: 4.2, [34]). According to their physical positions on the reference genome ARS1.2, the candidate genes were searched within the ±500 kb region using the biomaRt tool [35]. Then, the set of candidate genes was analyzed using the gene ontology (GO) and Kyoto Encyclopedia of Genes and Genomes (KEGG) pathway analyses using the Omicshare platform (https://www.omicshare.com/tools/, accessed on 2 February 2024).

## 3. Results

### 3.1. Phenotypic Analysis

For each of the eight body conformation traits (body height, body length, chest depth, chest width, chest girth, rump width, rump height, and cannon bone circumference), we calculated the arithmetic mean, standard deviation (SD), and coefficient of variation (CV) (Table 1). There were positive correlations between these eight traits (Figure 1B), with body height and rump height having the highest correlation coefficient (R = 0.91, *p* = 6.4 × 10^−61^), and body height and chest width the lowest (R = 0.60, *p* = 1.5 × 10^−16^). Based on the frequency distribution histogram and density trend line (Figure 1C–J), all body conformation traits were approximately normally distributed.

### 3.2. Whole-Genome Sequencing and Variant Calling

From the high-throughput sequencing of 100 Tashi goats, we obtained 1 676.4 Gb of raw data. On average, each sample was sequenced using approximately 60 million reads, and the Q20 (sequencing error rate < 0.01), Q30 (sequencing error rate < 0.001), and GC content values were 98.47%, 95.59%, and 41.8%, respectively. After alignment to the reference genome ARS1.2, the average mapping rate was above 99%, and the average sequencing depth was 6.2X. In this study, a total of 36,758,352 SNPs and 4,646,317 indels were obtained. The SNP distribution in all 29 goat chromosomes is shown in Appendix A. After quality control, 11 257 923 SNPs from 98 Tashi goats were retained for further analysis.

### 3.3. Population Structure Analysis

The PCA showed slight population stratification (Figure 2A). PC1 and PC2 accounted for 5.0% and 4.2% of the total variance, respectively. The major inflection points in the scree plots obtained from the PCA indicated two subpopulations. The relatedness coefficients of all individual pairs were between 0 and 0.1 (Figure 2B,C), suggesting no genetic relatedness among the samples. The population structural plots (Figure 2D,E) showed that the theoretical number of subpopulations (K) was two, based on cross-validation error values (0.637 at K = 2; 0.639 at K = 3). Individuals were divided into two subpopulations. These results showed that population structure and genetic relatedness should be included in the subsequent GWAS model.

### 3.4. Genome-Wide Association Study

Slight stratification was estimated by genomic inflation factors (body height, 1.03; body length, 1.02; chest depth, 1.04; chest width, 1.04; chest girth, 1.03; rump width, 0.99; rump height, 1.04; cannon bone circumference, 1.03). The QQ plots (Figure 3A–C) showed that most SNPs did not deviate from the expected *p*-values, suggesting that the models for the GWAS were reasonable. The distribution of association statistics for the GWAS is shown in the Manhattan plot (Figure 3A–D). For body height, we found 109 genome-wide significant SNPs (*p* < 1 × 10^−6^), and several strong signal peaks were found on chromosomes 7, 11, 24, and 29. For body length, 20 significant SNPs were located on chromosomes 2, 10, 11, 23, 24, and 29. For rump height, 70 significant SNPs were located on several chromosomes, with the strongest signal peak found on chromosome 11. For other body conformation traits, their corresponding Manhattan plot is shown in Appendix A. Based on the physical position of their significant SNPs, the closest annotated genes (located ±500 kb) were identified (Appendix A).

### 3.5. Identification of Core SNPs

Significantly correlated SNPs were found for all eight body conformation traits. However, considering that all eight traits are measures of body conformation, we integrated and compared eight Manhattan plots (Figure 4A). Of the 385 significant loci identified for all eight traits, the 109, 20, 52, 14, 62, 51, 70, and 7 SNPs were associated with body height, body length, chest depth, chest width, chest girth, rump width, rump height, and cannon bone circumference, respectively. Some SNPs were identified more than once: A total of 279 intersecting SNPs were distributed in different positions on 23 chromosomes. Trait values changed with SNP numbers (Figure 4B). For example, as SNP numbers increased, the body height of goats increased significantly (R = 0.76, *p* = 6.5 × 10^−20^); when SNP numbers increased from 0 to 10, the goats’ body length increased (R = 0.75, *p* = 1.1 × 10^−18^). These results illustrate the cumulative effects of correlated SNPs on these conformation traits.

Interestingly, we found several SNPs on chromosomes 10 and 11 that have been identified multiple times. The SNP chr11_89248159 occurred seven times and was not associated with chest width. The SNP chr11_89248733 occurred six times and was not associated with chest width or cannon bone circumference. Additionally, other adjacent SNPs occurred two (chr11_88833042, chr11_88840223, and chr11_89239285) or three (chr11_88216493, chr11_89214761, chr11_89242536, chr11_89248911, and chr11_89250659) times. On chromosome 10, six SNPs (chr10_25988403, chr10_26009053, chr10_26012238, chr10_26061262, chr10_26072624, chr10_26102739) within one 114 kb fragment occurred four or five times.

### 3.6. SNP Annotations and GO and KEGG Enrichment

The 500 kb region upstream and downstream of all 279 intersecting SNPs was annotated with 183 genes (Appendix A). The annotation via the KEGG pathway classified the annotated genes into 20 top KEGG pathways (Figure 5A). The most significantly enriched KEGG pathway was “olfactory transduction”. The enriched GO terms included “cellular process”, “biological regulation”, and “regulation of biological process”, belonging to Biological Process; “cellular anatomical entity” and “protein-containing complex”, belonging to Cellular Component; and “binding”, “catalytic activity”, and “molecular transducer activity”, belonging to Molecular Function (Figure 5B).

### 3.7. LD Analyses of Two Important Regions

Of the 279 intersecting SNPs, the 16 SNPs located in the above-mentioned fragments (chr.10:25988403-26102739, chr.11:88216493-89250659) were investigated further. In the 114 kb fragment on chromosome 10, six SNPs were located, and 14 related genes were annotated within ±500 kb (Table 2). Of these, chr10_25988403, chr10_26009053, and chr10_26012238 located the *FNTB* gene (from 25938275 to 26015218), and chr10_26072624 located the *CHURC1* gene (from 26062297 to 26078063). The six SNPs were tightly linked, and the TACTAG genotype was the highest at 91.8% (Figure 6A). In the 1.04 Mb fragment on chromosome 11, ten SNPs were located, and seven related genes were annotated within ±500 kb (Table 2). The LD analysis of ten polymorphisms showed several dependent LD blocks, whose SNPs were highly associated with one another (Figure 6B). Although these SNPs are located in intergenic regions, they may have regulatory potential for surrounding gene expression.

## 4. Discussion

The demand for goat meat is rising due to the ever-expanding human population and improvements in living standards [36]. For the genetic improvement of meat goats, goat growth and body conformation traits are critical factors in determining the productivity and profitability of goat farms [37]. This study explored the genetic variations influencing goats’ body conformation traits. By using whole-genome sequencing, we obtained millions of SNPs. Through the integration analysis of the Manhattan plot and LD heatmap, we found two regions located on chromosomes 10 and 11 that were strongly associated with body conformation traits. 

The body size of livestock directly impacts production performance, primarily in the following aspects: (1) Milk yield—Larger animals often produce more milk due to their larger size and greater amount of mammary gland tissue, giving them a higher production potential. (2) Cashmere or wool yield—Larger animals typically have more skin surface area, resulting in a larger effective area for skin production. Additionally, they have more hair follicles, producing more cashmere or wool. (3) Slaughter performance—Larger animals typically possess more muscle and fat, producing greater meat quality and carcass mass after butchering [38]. The body size of lambs at different ages directly affects carcass traits [39]. (4) Fertility—Healthy-sized animals typically exhibit better reproductive health, including lower rates of reproductive diseases and higher fertility rates. Normal development of reproductive organs is crucial for reproduction; excessive or insufficient size is detrimental to production. Juha Tuomi gives a generalized graph model of litter size to mammalian body size [40]. Xiao et al. explored and identified the correlation between 20 SNPs and body size and wool traits in alpine merino sheep [41]. John et al. performed a GWAS on reproductive traits and body conformation traits in Holstein cows [42], and Mohammed et al. performed a GWAS for calving performance and body conformation traits in Holstein cattle [43]. In meat goat breeds, prolificacy is highly related to body mass, parity, litter size, and body size [5].

Given the above discussion on several aspects of the effects of body size on livestock production, it is difficult to discover the genes related to body conformation traits at the genome level, especially using GWAS. As described before, the GWAS method has been used to mine the genetic architecture underlying goat body conformation traits [18,19]. In Russian Karachai goats, 241 SNPs and 238 candidate genes related to body conformation traits were identified, highlighting genes such as *CRADD*, *HMGA2*, and *MSRB3* for breeding improvements [19].

GWAS have been carried out for body size, not only for goats but also for other livestock. In sheep, 39 genes (including *FOSL2*, *KCND2*, *TGFBI*, *LECT2*, and *TRAK1*) were associated with body conformation traits, which could aid in marker-assisted selection in the sheep industry [44]. *KITLG* and *CADM2* were related to body height, and *MCTP1* and *COL4A6* were linked to chest circumference, which were further validated and shown to influence gene transcription activity [45]. In Yorkshire pigs, 11 candidate genes (*CDH13*, *SIL1*, *CDC14A*, *TMRPSS15*, *TRAPPC9*, *CTNND2*, *KDM6B*, *CHD3*, *MUC13*, *MAPK4*, and *HMGA1*) were associated with body traits, including body length, body height, chest circumference, abdominal circumference, cannon bone circumference, rump width, and chest width [46]. Wu et al. measured 29 body conformation traits of 1 314 Chinese Holstein cattle and discovered 59 genome-wide significant SNPs and several candidate genes (*DARC*, *GAS1*, *MTPN*, *HTR2A*, *ZNF521*, *PDIA6*, and *TMEM130*) [47]. In pigs, 60 significant genetic variants on chromosome 7 and their annotated genes (such as *INTS10*, *KIRREL3*, *SOX21*, *BMP2*, and *MAP4K3*) were associated with body conformation traits [48]. Zhang et al. identified 82 SNPs and candidate genes such as *BMP2*, *TNFAIP3*, *KDM4C*, and *HSPG2*, which were significantly associated with body conformation traits due to their role in growth and bone development [49].

Body conformation traits in domestic animals are complex quantitative traits controlled by multiple genes and influenced by the environment, as indicated by the inconsistent SNP positions and candidate genes identified in the above studies. In our study, 279 SNPs and 183 candidate genes were significantly correlated with body conformation traits. However, as with the studies listed above, these loci and candidate genes are newly associated with animal body conformation traits. Of these 279 SNPs and 183 genes, we identified important regions that occurred with high frequencies by integrating the GWAS results of all eight body conformation traits. We found two regions that fall on a narrow chromosomal interval on chromosomes 10 and 11, including 16 SNPs and 21 candidate genes. 

We know that the parameter ±500 kb, used for gene annotation based on the SNP position, is arbitrarily defined. For example, ±250 kb [50,51,52] and ±800 kb [53] have also been used for gene annotation in other studies. Therefore, we focused on the *FNTB* and *CHURC1* genes, where some significantly associated SNPs were located. The *FNTB* gene is a protein-coding gene that encodes for the beta subunit of the enzyme protein farnesyltransferase. Current research on this enzyme primarily focuses on its implications for human diseases. The polymorphisms in FNTB promoters are independent predictors of survival in patients with triple-negative breast cancer [54]. Increased expression of FNTB, along with Ras activation, leads to cardiomyocyte hypertrophy and facilitates apoptosis- and autophagy-driven cell death [55]. The *CHURC1* gene is a protein-coding gene. It plays a critical role in neural development processes and is involved in the regulation of various cellular mechanisms, but fewer studies have focused on this gene. Qiu et al. found an SNP (rs10138506) that influenced lung adenocarcinoma risk by affecting the *CHURC1* gene’s 3′ UTR length, potentially suppressing lung cancer [56]. *Metazoa_SRP* (Metazoan signal recognition particle RNA) and *RNF144A* (Ring Finger Protein 144A) are the genes closest to the most significant SNPs (chr11_89248159 and chr11_89248733). *Metazoa_SRP* is a signal-recognition particle essential for directing newly synthesized proteins to their correct cellular locations. The *RNF144* gene encodes a RING finger domain-containing E3 ubiquitin ligase and is associated with various types of cancer, including stomach, lung, breast, and ovarian cancers [57,58].

We identified many SNPs and candidate genes that may affect goats’ body conformation traits, some of which some genes have been reported in other studies. Body conformation traits of goats were strongly correlated with each other and shared similarities in the identified correlated SNP markers. Two critical regions (chr.10:25988403-26102739 and chr.11:88216493-89250659) and a series of candidate genes (such as *FNTB*, *CHURC1*, and *RNF144*) presented in this study are worth further exploration and could be used in molecular breeding of meat goats.

## 5. Conclusions

In this study, we measured eight body conformation traits in 155 adult Tashi goats and performed a whole-genome sequence on 100 goats. The individuals were divided into two subpopulations. All eight body conformation traits had a significantly positive correlation with each other, so their GWAS results had similar significant SNPs. We identified two regions (chr.10:25988403-26102739 and chr.11:88216493-89250659) and several candidate genes (such as *FNTB*, *CHURC1*, and *RNF144*) that were significantly associated with goat body conformation traits. Our results have important implications for understanding body conformation and for using molecular breeding in meat goats.

## Figures and Tables

**Figure 1 animals-14-01145-f001:**
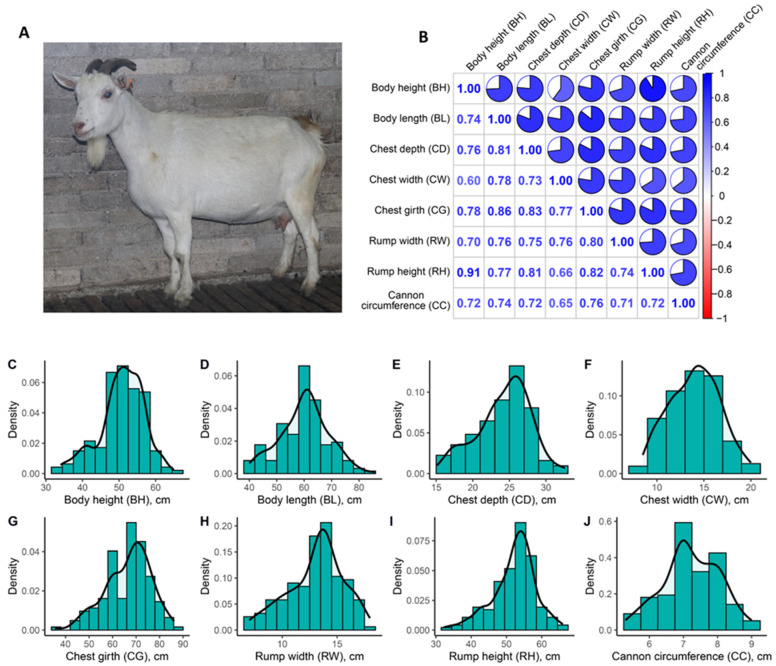
The goat breed and its body conformation traits. (**A**) A photo of an adult female Tashi goat; (**B**) A heatmap of the correlation coefficient between any two of eight body conformation traits; A trait frequency and distribution density line for body height (**C**), body length (**D**), chest depth (**E**), chest width (**F**), chest girth (**G**), rump width (**H**), rump height (**I**), and cannon bone circumference (**J**).

**Figure 2 animals-14-01145-f002:**
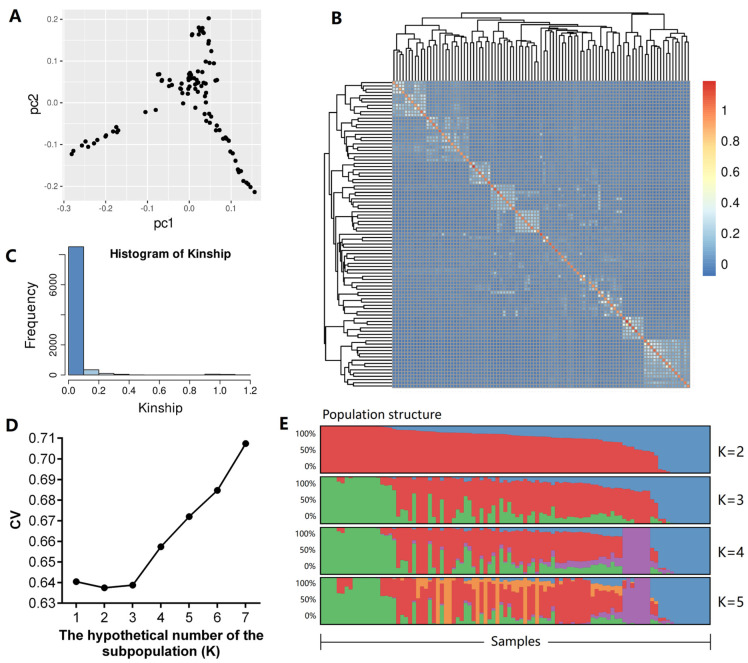
The structure analysis of the goat population. (**A**) A PCA score plot of all 100 goats based on genome sequence data; (**B**) A heatmap of the kinship matrix of all 100 goats; (**C**) A histogram frequency distribution of pairwise relative kinship coefficients; (**D**) The line chart of cross-validation error changes with the number of hypothetical subpopulations; (**E**) A barplot of the structure proportion.

**Figure 3 animals-14-01145-f003:**
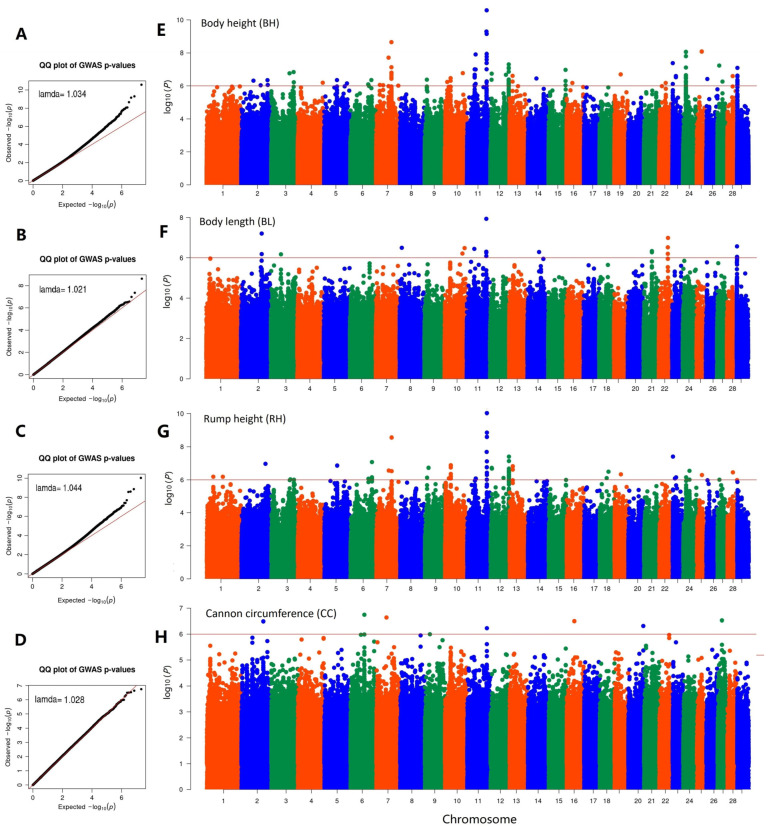
Manhattan plots of four body conformation traits in Tashi goats. QQ plots of GWAS results for body height (**A**), body length (**B**), rump height (**C**), and cannon bone circumference (**D**); Manhattan plots of GWAS results for body height (**E**), body length (**F**), rump height (**G**), and cannon bone circumference (**H**).

**Figure 4 animals-14-01145-f004:**
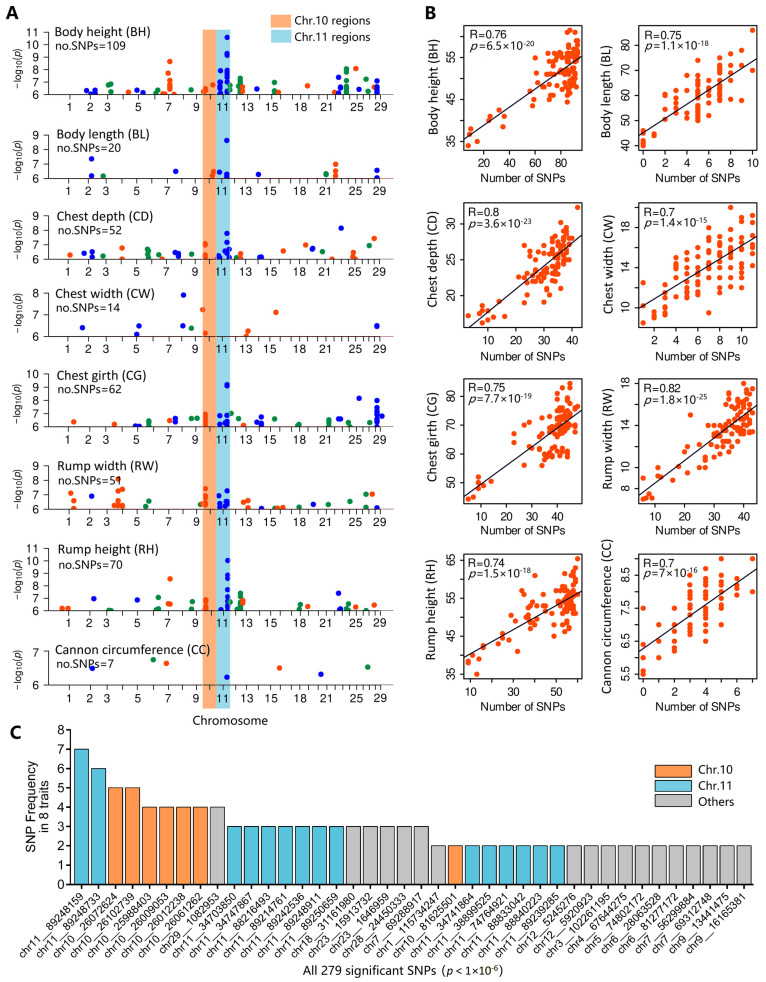
Significantly correlated SNPs of eight body conformation traits. (**A**) The overlapping comparison of the portions above the threshold line (1 × 10^−6^) for eight Manhattan plots; (**B**) The trait values changed with the number of significant SNPs in corresponding Manhattan plots for body height, body length, chest depth, chest width, chest girth, rump width, rump height, and cannon bone circumference. (**C**) The frequency of SNPs in all eight Manhattan plots.

**Figure 5 animals-14-01145-f005:**
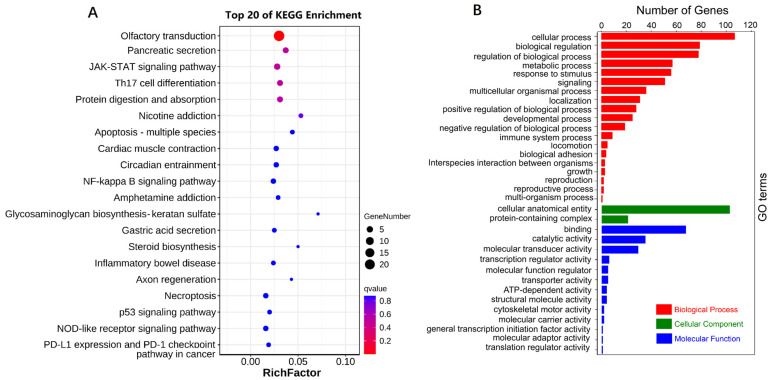
Function enrichment of 183 related genes. (**A**) A bubble diagram of KEGG pathways; (**B**) A barplot of the enriched GO terms.

**Figure 6 animals-14-01145-f006:**
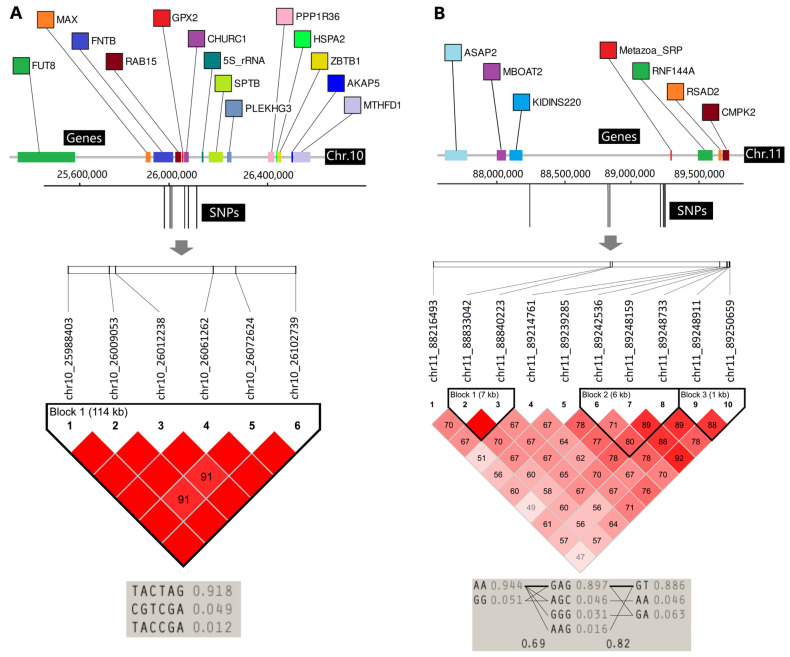
A graphical representation of the chromosome structure and core SNP locations. (**A**) Six SNPs in a 114 kb fragment were annotated with 14 related genes and tightly locked; (**B**) Ten SNPs in a 1.03 Mb fragment were annotated with six related genes, and three LD blocks were identified.

**Table 1 animals-14-01145-t001:** Descriptive statistics of eight body conformation traits (*n* = 155).

Item	Body Height	Body Length	Chest Depth	Chest Width	Chest Girth	Rump Width	Rump Height	Cannon Bone Circumference
	(BH)	(BL)	(CD)	(CW)	(CG)	(RW)	(RH)	(CC)
Mean	50.73	60.09	24.20	13.99	66.60	13.09	52.14	7.22
Std	6.08	8.84	3.65	2.62	9.57	2.49	6.17	0.82
Max	65.00	86.00	32.60	20.00	86.20	18.00	66.00	9.00
Min	34.00	40.00	15.10	8.50	35.50	7.00	32.20	5.30
CV	12.0%	14.7%	15.1%	18.7%	14.4%	19.0%	11.8%	11.4%

Std: standard deviation; CV: coefficient of variation.

**Table 2 animals-14-01145-t002:** The significantly correlated SNPs located in two regions and their related genes.

Interval No.	Interval Length	SNP Maker	Frequency	Chromosome	Position	Mean of PVE	Genes (±500 kb)
1	114 Kb	chr10_25988403	4	chr10	25,988,403	22.41%	*FUT8*, *MAX*, *FNTB*, *RAB15*, *GPX2*, *CHURC1*, *5S_rRNA*, *SPTB*, *PLEKHG3*, *PPP1R36*, *HSPA2*, *ZBTB1*, *AKAP5*, *MTHFD1*
		chr10_26009053	4	chr10	26,009,053	21.64%	
		chr10_26012238	4	chr10	26,012,238	25.21%	
		chr10_26061262	4	chr10	26,061,262	22.73%	
		chr10_26072624	5	chr10	26,072,624	24.78%	
		chr10_26102739	5	chr10	26,102,739	26.30%	
2	1.03 Mb	chr11_88216493	3	chr11	88,216,493	20.81%	*ASAP2*, *MBOAT2*, *KIDINS220*, *Metazoa_SRP*, *RNF144A*, *RSAD2*, *CMPK*
		chr11_88833042	2	chr11	88,833,042	20.54%	
		chr11_88840223	2	chr11	88,840,223	20.77%	
		chr11_89214761	3	chr11	89,214,761	20.94%	
		chr11_89239285	2	chr11	89,239,285	20.81%	
		chr11_89242536	3	chr11	89,242,536	21.35%	
		chr11_89248159	7	chr11	89,248,159	15.45%	
		chr11_89248733	6	chr11	89,248,733	28.40%	
		chr11_89248911	3	chr11	89,248,911	20.18%	
		chr11_89250659	3	chr11	89,250,659	8.65%	

Note: Frequency refers to the number of occurrences in all eight traits; PVE refers to the percentage of phenotypic variation explained for each significant SNP; the “mean of PVE” is the arithmetic mean of PVE values for all eight body conformation traits.

## Data Availability

The sequencing data included in this study is available at NCBI BioProject PRJNA1078072.

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
