# Peer review of "Genome-Wide Association Study of Body Conformation Traits in Tashi Goats (*Capra hircus*)"

_animals, 2024, doi:10.3390/ani14081145_

Round 1

Reviewer 1 Report

Comments and Suggestions for Authors

The manuscript is well written and results clearly presented.

The authors should the variance explained for each significant SNP and show which proportion of the phenotypic is explained for each trait ba the significant SNPs.

Traits are phenotypically highly correlated.

Can the authors show the correlations amongs significant SNPs.

Another point may be that the authors calculate PC for body traits and explore the significants SNPs. This may be useful to explain the most important SNPs for body growth in a more general way.

Comments on the Quality of English Language

No comments

Author Response

reviewer-1-author-notes

1. The manuscript is well written and results clearly presented.
Response: 
Thanks for your positive comment on our work.

2. The authors should the variance explained for each significant SNP and show which proportion of the phenotypic is explained for each trait ba the significant SNPs.
Response: 
(1) Thanks for this point. We have calculated the PVE (the percentage of phenotypic variation explained) for each significant SNP. 
(2) Please see Table 2; the “mean of PVE” column was added.

3. Traits are phenotypically highly correlated.
Can the authors show the correlations amongs significant SNPs.
Response: 
(1) The eight traits are phenotypically highly correlated; therefore, the GWAS relsults was similar. 
(2) The significant SNPs for each trait had a high correlation, and we used the frequency diagram (Figure 4C is actually a Venn plot, showing SNPs co-identified significant SNPs for eight traits) to measure this correlation.

4. Another point may be that the authors calculate PC for body traits and explore the significants SNPs. This may be useful to explain the most important SNPs for body growth in a more general way.
Response: 
Thanks for your positive comment; we added Figure S1 (The PCA score plots based on phenotypical data). 

Reviewer 2 Report

Comments and Suggestions for Authors

Dear Authors,

I have a main question: why did you measure phenotypes (and report results about that) in 155 goats and then you genotyped only 100? If I understood correctly and you genotyped only 100 goats, you must report phenotypes only for the 100 animals with genotypes.

I have also a main concern about the model: why you did not consider the farm (as random effect) and the age of the animal (as fixed covariate)?

Lines 17-18 and 33-34: You analyzed 8 traits, and you reported eight values of significant SNPs. However, you listed just seven phenotypes at lines 18 and 34, body height (1), length (2), chest depth (3), chest width (4), chest girth (5), rump height (6), and cannon bone circumference (7).

Line 33: “related.The” a space is missing; moreover, you may consider “A total of” instead of “The”.

Line 35 and lines 239-240: Why did you consider “Within the ± 500 kb region”?

Line 68: “Maria et al. genotyped” then you use reference [8]. The last name of the first author of reference 8 is Luigi-Serra, so it should be “Luigi-Serra et al. genotyped…”

Line 77: “… analyzing correlations between …” is probably better to say “… analyzing associations between …”

Line 82: “SNPs.The” a space is missing.

Line 96: “ap-proved” should be “approved”

Line 139: “using bwa” is this a software? If yes, please write “using bwa software”.

Line 165: Did you consider farm and age as effects in the model? I think that the age of the animal has a strong effect on the morphological traits.

Line 169: Did you really use this (P < 10-6) threshold?

Line 222: Why did you perform PCA and relationship matrix only on 100 goats?

Lines 231-232: “suggesting that the models for the GWAS were reasonable” any reference to support this sentence?

Line 234: “… significant SNPs (P < 1e-6),” in M&M, at line 169, you wrote that significance was declared for P<10-6. Please clarify.

Line 290: The names of the GO terms listed in Figure 5B are impossible to read.

Lines 379-380: “We know that the parameter ± 500 kb, used for gene annotation based on the SNP position, is arbitrarily defined” I commend the authors for recognize this. You may discuss this threshold with other studies using similar values (e.g., 10.9787/PBB.2022.10.2.139; 10.3390/ani10081300; 10.1371/journal.pone.0257461; 10.3390/ijms25031756).

Comments on the Quality of English Language

No particular problems with English.

Author Response

reviewer-2 -author-notes

1.I have a main question: why did you measure phenotypes (and report results about that) in 155 goats and then you genotyped only 100? If I understood correctly and you genotyped only 100 goats, you must report phenotypes only for the 100 animals with genotypes.
Response: 
(1)In the phenotypic investigation, we measured traits in 155 Tashi goats. Due to the limited project funding, we only tested the genome sequencing of 100 goats.
(2)We thought that the sample size from the phenotypic and genomic investigation stages did not need consistency.

2.I have also a main concern about the model: why you did not consider the farm (as random effect) and the age of the animal (as fixed covariate)?
Response: 
(1)Tashi goat is a national-level goat breed, also called the “Guizhou white goat”.
(2)There are no intensive breeding farms in the land where Tashi goats belong. Because these goats come from many farmers (the goat number raised by each farmer is not large), we did not consider “farm” as a random effect.
(3)The “sex” and “age” are the fixed effects, line 164.

3.Lines 17-18 and 33-34: You analyzed 8 traits, and you reported eight values of significant SNPs. However, you listed just seven phenotypes at lines 18 and 34, body height (1), length (2), chest depth (3), chest width (4), chest girth (5), rump height (6), and cannon bone circumference (7).
Response: 
Thanks for this point. The trait “rump width” was added, and we have revised the sentence as “body height, body length, chest depth, chest width, chest girth, rump width, rump height, and cannon bone circumference”.

4.Line 33: “related.The” a space is missing; moreover, you may consider “A total of” instead of “The”.
Response: 
Thanks for your point; we have revised this error.

5.Line 35 and lines 239-240: Why did you consider “Within the ± 500 kb region”?
Response: 
(1) We set the pamarter “within the ± 500 kb region” based on other reference papers.
(2) No matter how large the interval is set, it does not affect the conclusion of this paper, because the core SNPs fall on the few genes we focused.

6.Line 68: “Maria et al. genotyped” then you use reference [8]. The last name of the first author of reference 8 is Luigi-Serra, so it should be “Luigi-Serra et al. genotyped…”
Response: 
Thanks for your point; we have revised this error.

7.Line 77: “… analyzing correlations between …” is probably better to say “… analyzing associations between …”
Response: 
Thanks for your point; we have revised this word.

8.Line 82: “SNPs.The” a space is missing.
Response: 
Thanks for your point; we have revised this error.

9.Line 96: “ap-proved” should be “approved”
Response: 
Thanks for your point; we have revised this word.

10.Line 139: “using bwa” is this a software? If yes, please write “using bwa software”.
Response: 
Thanks for your point; we have revised it as “using bwa software ” according to your advice.

11.Line 165: Did you consider farm and age as effects in the model? I think that the age of the animal has a strong effect on the morphological traits.
Response: 
(1)As responded to in comment 2, we did not consider “farm” because this factor is difficult to define.
(2)Although the animals are adult goats, the factor “age” was considered as one fixed effect.

12.Line 169: Did you really use this (P < 10-6) threshold?
Response: Yes, we strictly enforced the SNP significance threshold of P<10e-6.

13.Line 222: Why did you perform PCA and relationship matrix only on 100 goats?
Response: 
(1)The PCA and relationship matrix were calculated based on genome sequence data; because we only sequenced 100 goats.
(2)In this revision stage, we performed PCA analysis based on phenological data, as shown in Figure S1.

14.Lines 231-232: “suggesting that the models for the GWAS were reasonable” any reference to support this sentence?
Response: 
According to the QQ plots, most SNPs had good consistency between observed- and expected- P-values; besides, the genomic inflation factor (lamda) was within a normal range of 0.95 to 1.05. Therefore, the models for the GWAS were reasonable.

15.Line 234: “… significant SNPs (P < 1e-6),” in M&M, at line 169, you wrote that significance was declared for P<10-6. Please clarify.
Response: 
Thanks for your point; we have uniformly described as 10e-6.

16.Line 290: The names of the GO terms listed in Figure 5B are impossible to read.
Response: 
Thanks for your point; we have redrawn Figure 5B, and the GO terms can be easily read now.

17.Lines 379-380: “We know that the parameter ± 500 kb, used for gene annotation based on the SNP position, is arbitrarily defined” I commend the authors for recognize this. You may discuss this threshold with other studies using similar values (e.g., 10.9787/PBB.2022.10.2.139; 10.3390/ani10081300; 10.1371/journal.pone.0257461; 10.3390/ijms25031756).
Response: 
Thanks for your suggestion. The papers were added to the discussion section in line 367.

Reviewer 3 Report

Comments and Suggestions for Authors

I would suggest to the authors to provide more information about the selection of the goats which were included in the study. It is only written that the goats were randomly selected. Which was the total number of goats’ form which the 155 Tashi goats were randomly selected? Which was the proof that they belong to Tashi breed? Are the Tashi goats recorded in a National Register of Tashi breed?

Why from a total of 155 goats you used 100 for GWARS? Please provide further information in discussion section.

Please also read the manuscript review included in the attached file 

Comments on the Quality of English Language

Please review the minor spelling errors from the entire manuscript

Author Response

reviewer-3-author-notes

1.The aim of the paper is to study the genomic variations influencing body conformation traits, by performing a genome-wide association study on Tashi goats (an indigenous Chinese goat breed). In order to assess the body conformation traits, a total of 155 Tashi goats were phenotyped for eight body conformation traits, as follows: body height, body length, chest depth, chest width, chest girth, rump width, rump height, and cannon bone circumference. The study was made on a purpose to identify the genetic markers of economically valuable traits which can have practical benefits for the meat goat industry.
Response: 
Thanks for your positive comment on our work.

2.The scientific content of the article is very high, being obtained 1 676.4 Gb of raw data and identified 11 257 923 qualified single nucleotide polymorphisms (SNPs). GWAS revealed 109, 20, 52, 14, 62, 51, 70, and 7 SNPs significantly associated with body height, length, chest depth, chest width, chest girth, rump height, and cannon bone circumference, respectively. Based on the significant SNPs' physical locations 183 genes were annotated. Notably, several SNPs have been identified multiple times in two regions, chr.10:25988403- 26102739 and chr.11:88216493-89250659, and the following candidate genes, such as FNTB, CHURC1, and RNF144, were identified as possible being crucial for goat body conformation traits.
Response: 
Thanks for your positive comment on our work.

3.The applied methodology is at a high scientific level in the field.
Response: 
Thanks for your positive comment on our work.

4.The materials and methods chapter includes very modern and up-to-date molecular genetics methods and statistics.
Response: 
Thanks for your positive comment on our work.

5.I would suggest to the authors to provide more information about the selection of the goats which were included in the study. It is only written that the goats were randomly selected. Which was the total number of goats’ form which the 155 Tashi goats were randomly selected? Which was the proof that they belong to Tashi breed? Are the Tashi goats recorded in a National Register of Tashi breed? Why from a total of 155 goats you used 100 for GWARS? Please provide further information in discussion section.
Response: 
(1)Tashi goat is a national-level goat breed, also called the “Guizhou white goat”. But the Tashi breed is generally considered a subtype of “Guizhou white goat”.
(2)The Tashi goats only have ~10000 animals in the 2022 Genetic Resource Investigating of China. 
(3)Limited to research funding, we could only afford to sequence 100 Tashi goats (average sequencing depth was 6.2X). We added some sentences to explain; please see line 126.

6.The reference list is exhaustive, and includes the most important publication in the field.
Response: 
Thanks for your positive comment on our work.

7.The manuscript is well-structured and the discussions are comparatively emphasizing the obtained results and the scientific knowledge in the field.
Response: 
Thanks for your positive comment on our work.

8.The results are well presented, together with the graphics which better represents the obtained results, also complemented by the supplementary files.
Response: 
Thanks for your positive comment on our work.

9.I would suggest to the authors to focus more in the discussion section to discuss the reasons why two subpopulations of Tashi goats were identified.
Response: 
Thanks for this suggestion. After careful consideration, we do not add some discussion on “two subpopulations” for the following reasons:
(1)Tashi goat breed is a meat-type goat with a small body size, which belongs to the Guizhou white goat in animal resource management. In this study, we found that this breed can be divided into two subpopulations, but it is not significant; it may be due to the diversity of sample collection sites.
(2)We are carrying out a larger genomic survey, which includes other goat breeds, including Guizhou black goat, Guizhou horse-head goat and so on. This project will enable a more comprehensive analysis of these goat breeds' genetic evolution, population structure and relationships.

10.The final conclusions are based on the obtained results, are well presented and synthetized.
Response: 
Thanks for your positive comment on our work.

11.The animal study protocol was approved by the Guizhou University Experimental Animal Ethics Committee). (Number:EAE-GZU-2023-E047). I will also suggest the authors to correct the minor spelling errors in the original text.
Response: 
(1)To the sentence your point, we have revised it as “This study was approved by the Guizhou University Experimental Animal Ethics Committee (No. EAE-GZU-2023-E047)”.
(2)We have carefully checked and revised the manuscript accordingly.

Round 2

Reviewer 2 Report

Comments and Suggestions for Authors

Dear authors,

I do not have other questions for you.

Comments on the Quality of English Language

no comments